# In vivo targeting of de novo DNA methylation by histone modifications in yeast and mouse

**Marco Morselli[1]\*, William A Pastor[1], Barbara Montanini[2], Kevin Nee[1], Roberto Ferrari[1], Kai Fu[1], Giancarlo Bonora[1,3], Liudmilla Rubbi[1], Amander T Clark[1], Simone Ottonello[2], Steven E Jacobsen[1,3,4]\*, Matteo Pellegrini[1,3]\***

[1]Department of Molecular, Cell and Developmental Biology, University of California, Los Angeles, Los Angeles, United States; [2]Biochemistry and Molecular Biology Unit, Department of Life Sciences, Laboratory of Functional Genomics and Protein Engineering, Parma, Italy; [3]Eli and Edythe Broad Center of Regenerative Medicine and Stem Cell Research, University of California, Los Angeles, Los Angeles, United States; [4]Howard Hughes Medical Institute, University of California, Los Angeles, Los Angeles, United States

**Abstract** Methylation of cytosines ($5^{me}C$) is a widespread heritable DNA modification. During mammalian development, two global demethylation events are followed by waves of de novo DNA methylation. In vivo mechanisms of DNA methylation establishment are largely uncharacterized. Here, we use *Saccharomyces cerevisiae* as a system lacking DNA methylation to define the chromatin features influencing the activity of the murine DNMT3B. Our data demonstrate that DNMT3B and H3K4 methylation are mutually exclusive and that DNMT3B is co-localized with H3K36 methylated regions. In support of this observation, DNA methylation analysis in yeast strains without Set1 and Set2 shows an increase of relative $5^{me}C$ levels at the transcription start site and a decrease in the gene-body, respectively. We extend our observation to the murine male germline, where H3K4me3 is strongly anti-correlated while H3K36me3 correlates with accelerated DNA methylation. These results show the importance of H3K36 methylation for gene-body DNA methylation in vivo.

\*For correspondence:
mmorselli@ucla.edu (MM);
jacobsen@ucla.edu (SEJ);
matteop@mcdb.ucla.edu (MP)

**Competing interests:** The authors declare that no competing interests exist.

**Reviewing editor**: Bing Ren, University of California, San Diego School of Medicine, United States

## Introduction

In multicellular organisms, every cell type possesses the same genetic information, but manifests a different phenotype. Chromatin plays a fundamental role in both the establishment and maintenance of each cell's state. Many players contribute to chromatin states, including nucleosome organization, histone post-translational modifications, and non-coding RNAs (*Chen and Dent, 2014*; *Maze et al., 2014*; *Quinodoz and Guttman, 2014*). Another mechanism for maintaining the state of a cell through cell division is the methylation of cytosines at position 5 ($5^{me}C$), a widespread heritable DNA modification found in prokaryotes, plants, several fungi, and animals (*Iyer et al., 2011*). In mammals, DNA methylation plays a fundamental role in processes such as imprinting, X-chromosome inactivation, transposon inactivation, and gene expression regulation (*Smith and Meissner, 2013*). Dysregulation of DNA methylation is a common feature in cancer (*Eden et al., 2003*; *You and Jones, 2012*) and a variety of human diseases are caused by defective imprinting (*Peters, 2014*).

Methylation is mainly found at symmetric CpG dinucleotides, where it is introduced by the de novo DNA methyltransferases (DNMT3a and DNMT3b) and can be copied faithfully during DNA replication by the activity of a 'maintenance' DNA methyltransferase, DNMT1 (*Law and Jacobsen, 2010*). However, DNA methylation is not static throughout mammalian development. In fact, $5^{me}C$ can either

**eLife digest** In animals and other multicellular organisms, there are many different types of cells that each perform particular roles in the body. This is possible because the genetic information—which is the same in all cells—is controlled so that only a subset of all the genes within an individual cell are 'switched on' at a particular time.

Genetic information is contained within molecules of DNA, which are wrapped around proteins called histones. The genes in regions of DNA where these histones are packed tightly together tend to be switched off, while genes in regions of DNA that are loosely packed tend to be switched on. The level of packaging is controlled by the addition of 'methyl' tags to the histone proteins.

These tags can also be added directly to the DNA in a process called DNA methylation. Enzymes called methyltransferases add the tags to the DNA, which tends to switch off the gene. The locations of the methyl tags can be copied when the DNA replicates before the cell divides so that the pattern of DNA methylation can be passed on to its daughter cells. However, it is not clear how the methyltransferases are able to target particular regions for methylation.

To address this question, Morselli et al. introduced a methyltransferase called DNMT3b into yeast, a single-celled organism that does not normally add methyl tags to its DNA. The experiments show that the activity of the enzyme is affected by the presence of methyl tags on certain histone proteins. For example, a methyl tag at one particular site on a histone, called H3K4, prevents the DNMT3b enzyme from adding methyl tags to DNA. However, a methyl tag at another site called H3K36 promotes DNA methylation.

Morselli et al. found that these two histone sites had similar effects on DNA methylation in mouse sperm cells. Morselli et al.'s findings may be useful in the future development of treatments for cancer and other diseases that are caused by defects in DNA methylation.

be lost by a passive mechanism, such as the failure to maintain DNA methylation through cell division or by an active mechanism such as the removal of methylcytosine, typically via an oxidized intermediate (*Pastor et al., 2013*).

Demethylation and de novo methylation can occur in a locus-specific manner, typically in concert with the activation or silencing of promoters or enhancers. However, global demethylation and de novo methylation events can also occur during development (*Pastor et al., 2013*; *Seisenberger et al., 2013*). For example, most DNA methylation is progressively lost between fertilization and the formation of the blastula and global de novo DNA methylation then occurs coincidently with implantation of the embryo. This de novo methylation event largely shapes the methylation pattern of the animal, with additional changes occurring in somatic tissues, which contribute to cellular identity. In the germline however, a second reprogramming event occurs. After specification of the germ cells, most DNA methylation is lost during early primordial germ cell (PGCs) development. Unlike in early embryogenesis, imprints are erased during this period. Genome-wide de novo methylation then occurs before birth in the male germline and upon oocyte maturation in females (*Smallwood et al., 2011*). This de novo methylation event establishes the imprints that are inherited in the next generation.

Considering the importance of local and global de novo methylation events in imprinting, gene regulation and cellular identity, it is important to understand how the de novo DNA methyltransferases are targeted to the correct genomic regions. DNMT3 proteins do not have strong sequence preferences beyond CpG dinucleotides (*Dodge et al., 2002*). We therefore sought to determine which factors are critical for the targeting of de novo DNA methyltransferases.

Active de novo DNA methyltransferases possess three different domains: the catalytic domain, found at the C-terminus of the protein, an ADD domain and a PWWP domain (*Figure 1A*) (*Law and Jacobsen, 2010*). In contrast, the inactive DNMT3L possesses only a functional ADD domain. The ADD domains of all three DNMTs have been shown to preferentially bind histone 3 tails that lack methylation at lysine 4 (H3K4me0) (*Ooi et al., 2007*; *Zhang et al., 2010*), and this binding has been recently shown to relieve DNMT3a auto-inhibition (*Guo et al., 2015*). This is consistent with the observation that genomic regions bearing H3K4 methylation are generally depleted of 5$^{me}$C (*Singh et al., 2013*). The PWWP domain of several proteins has been shown to bind H3K36

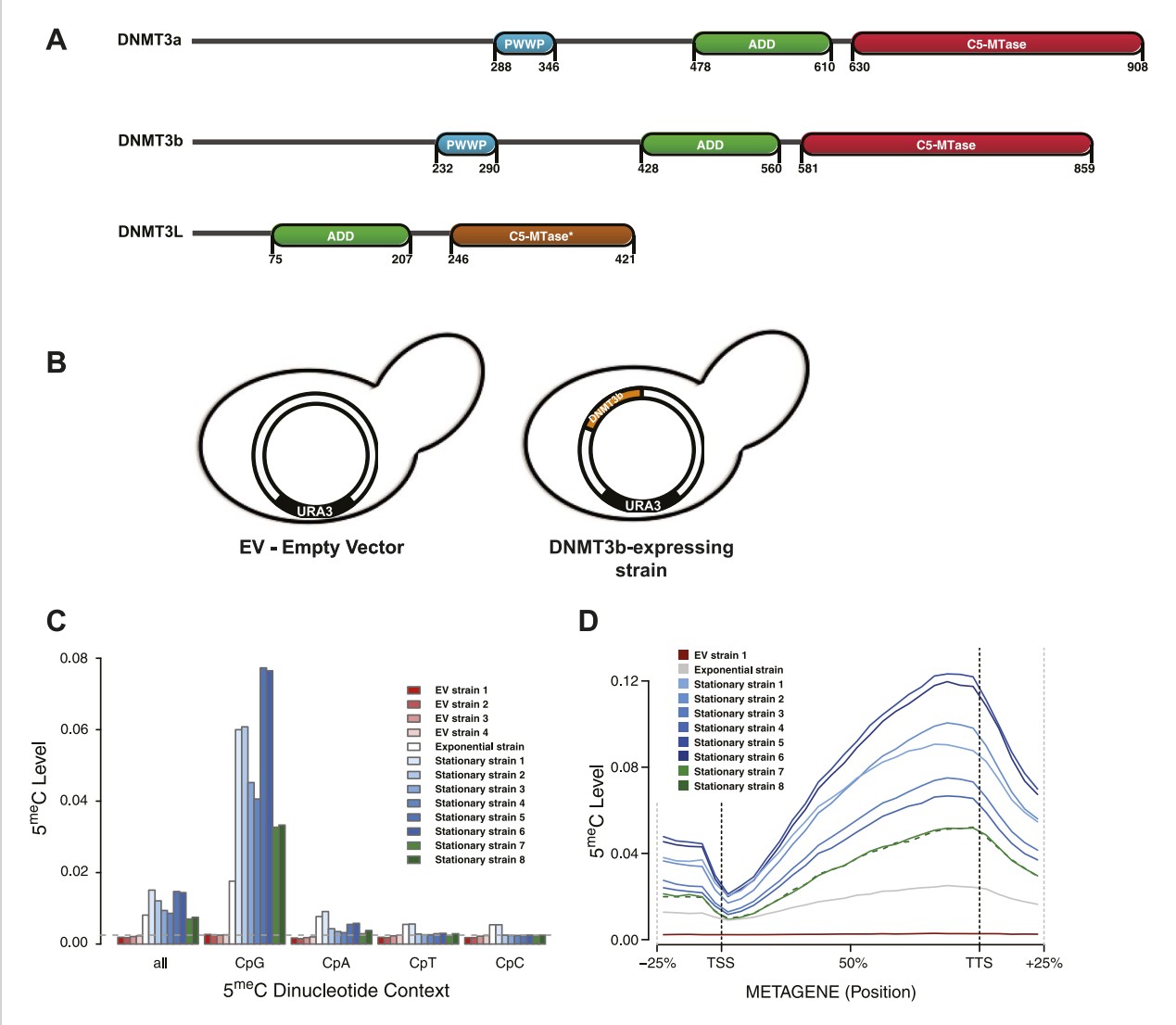

**Figure 1**. Distribution of induced DNA methylation in *Saccharomyces cerevisiae*. (**A**) Murine DNMT3 proteins with known domains: PWWP, ADD (ATRX–DNMT3–DNMT3L), and C-5 methyltransferase domain (not functional in DNMT3L). Accession numbers: DNMT3a = O88508; DNMT3b = O88509; DNMT3L = Q9CWR8. (**B**) Constructs used in this study. The empty vector (EV) is pYES2 (Life Technologies). DNMT3b expression is controlled by the GAL1 promoter. (**C**) Levels of 5meC in different dinucleotide contexts. The gray dotted line represents the unconversion rate. (**D**) Metagene plot of CpG methylation in cells expressing DNMT3b during logarithmic and stationary phase. EV (strain not expressing DNMT3b). Exponential and stationary strains 1–6 are derived from the W303 strain, while stationary strains 7 and 8 are in a BY4741 background.

The following figure supplements are available for figure 1:

**Figure supplement 1**. Chromosome-wide view of DNA methylation and genomic features.

**Figure supplement 2**. Distribution of 5meC around TSS and TTS.

methylation (*Vermeulen et al., 2010*), and indeed the DNMT3a-PWWP domain has also been shown to interact with the tri-methylated lysine 36 of histone H3 (H3K36me3) in vitro (*Dhayalan et al., 2010*). The importance of these histone-binding domains in targeting DNA methyltransferase activity in vivo is still unclear. It is also possible that the PWWP domain's primary function is to bind DNA and not nucleosomes (*Dhayalan et al., 2010*). Recently, it has been reported that the PWWP domain is important in specifying the localization of DNMT3b in mouse embryonic stem cells (*Baubec et al., 2015*).

While there has been extensive characterization of DNMT3 in vitro, a comprehensive analysis of the mechanisms guiding the activity of a de novo DNMT in vivo is still incomplete. To address this question, we introduced DNMT3b into an organism that has no endogenous DNA methylation machinery, the budding yeast *Saccharomyces cerevisiae*, to study the chromatin components affecting the activity of a mammalian de novo DNA methyltransferase. This system has several advantages over the study of DNA methylation in mammalian cells. Yeast has conserved histone sequences and many residues are modified at the same sites as those found in higher eukaryotes. However, unlike mammalian cells, yeast cells can be easily manipulated and the small size of their genome reduces costs associated with next-generation sequencing-based approaches. Moreover, yeast has already been used to show the importance of the N-terminus of histone H3 in targeting the DNA methylation complex (*Hu et al., 2009*).

Our data show that the chromatin template guides the activity of DNMT3b. DNMT3b preferentially deposits methylation in linker DNA compared to nucleosomal DNA. Also, DNMT3b activity correlates positively with H3K36me3 and negatively with H3K4me3. In fact, mutation of the H3K36 methyltransferase Set2 decreases DNA methylation over regions that would normally contain H3K36me3. Thus the marks themselves, as opposed to genomic features that correlate with these marks, are responsible for targeting DNA methylation. We also demonstrate that the pattern of H3K4 and H3K36 methylation in embryonic male germ cells accurately predicts which regions undergo de novo methylation, indicating that the mechanism observed in yeast is conserved in mammals.

## Results

### Ectopically expressed DNMT3b methylates yeast genomic DNA

*S. cerevisiae* does not have any endogenous cytosine DNA methyltransferases, and its DNA is therefore unmethylated. To study the activity of a de novo methyltransferase in this organism, we introduced the murine DNMT3b under the control of the inducible GAL1 promoter (*Figure 1B*). We measured the levels of 5-methylcytosine ($5^{me}C$) in these strains using whole genome bisulfite sequencing (WGBS) (*Supplementary file 1A*). We observed significant levels of $5^{me}C$ of DNA extracted from the exponentially growing and stationary phases of the same strain culture (*Figure 1C* and *Supplementary file 2A*), with higher methylation levels observed in stationary phase. CpG dinucleotides were preferentially methylated, as expected from the previously characterized activity of mammalian DNMT3. The methylation levels of CpG dinucleotides range from 3.3 to 7.7%, depending on the yeast strain analyzed. These levels are about 10–20 times higher than the average of other dinucleotides levels (*Supplementary file 2A*), and well above the bisulfite non-conversion rate of 0.27%, as estimated from an unmethylated lambda DNA spike-in.

Despite some level of variability, we observe methylation across the entire yeast genome (*Figure 1—figure supplement 1A,B*). When mapping reads to the genome we only retain those that map to a single position. As a result we do not obtain methylation estimates for regions that contain repetitive sequences, such as the rRNA containing regions in chromosome XII.

We also observed a striking methylation distribution within genes (*Figure 1D*), with low levels at the transcription start site (TSS) and increasing methylation in the gene body, reaching a maximum close to the transcription termination site (TTS). The same pattern is found in mammals (*Lister et al., 2009*; *Chodavarapu et al., 2010*), suggesting that equivalent mechanisms regulating DNMT3 activity in mammalian genes might also be present in yeast.

### DNMT3b preferentially methylates linker DNA

In yeast, nucleosomes are well positioned at the beginning of a gene, with nucleosome-free regions (NFRs) immediately upstream of the TSS and downstream of the TTS (*Brogaard et al., 2012*). When average levels of $5^{me}C$ are calculated around the TSS, we observed a periodicity of about 170 bp (*Figure 1—figure supplement 2*). A similar periodicity is also observed at the TTS. This suggested that nucleosomes might influence the activity of de novo DNMTs.

To address this question, we measured nucleosome positioning genome-wide using micrococcal nuclease-digested chromatin and deep-sequencing (MNase-seq) (*Supplementary file 1B* and *Supplementary file 3A,B*). We profiled the distribution of methylated cytosines at the TSS (*Figure 2A*), TTS (*Figure 2B*), and around each nucleosome center (*Figure 2C*).

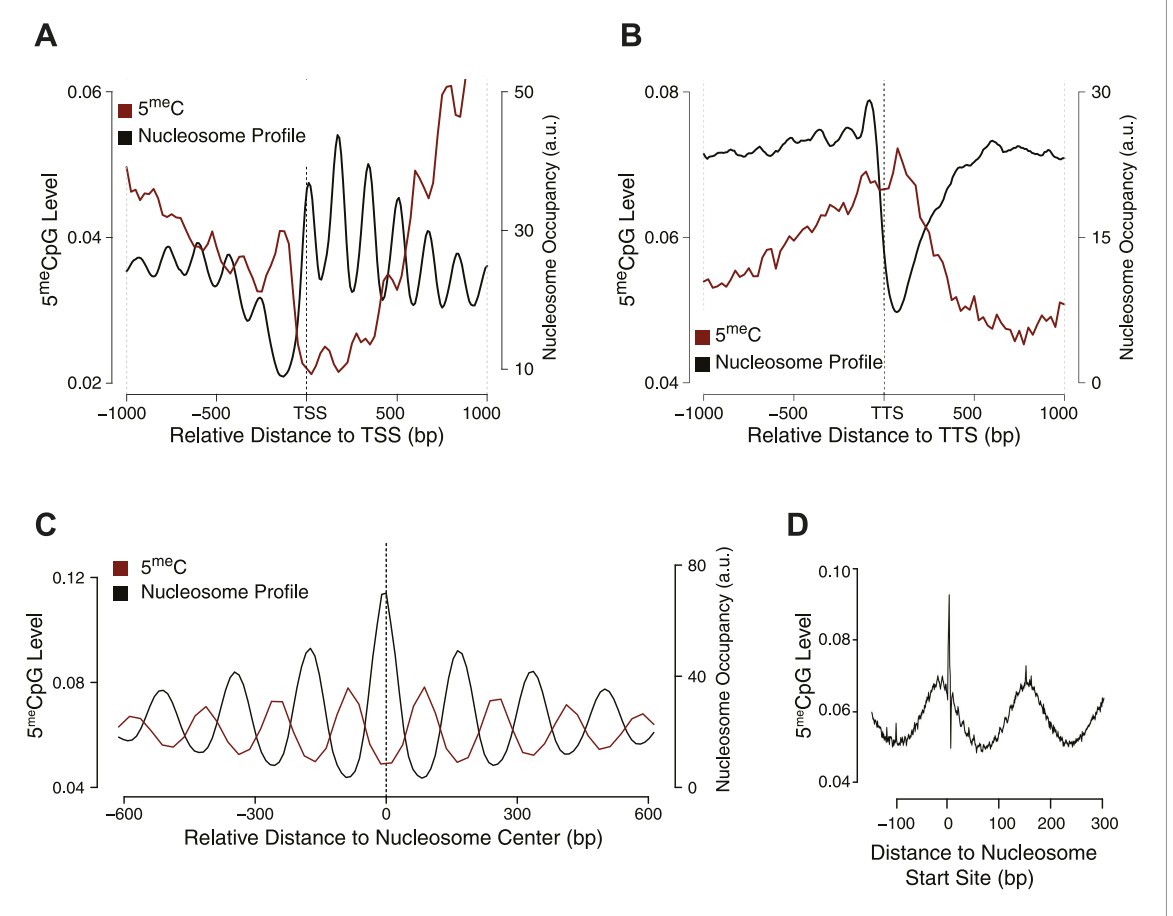

**Figure 2**. Influence of nucleosome positioning on DNA methylation. Average distribution of nucleosomes and DNA methylation (CpG context) around (**A**) Transcriptional Start Site (TSS), (**B**) Transcriptional Termination Site (TTS), and (**C**) nucleosome centers. (**D**) Meta-nucleosome plot of CpG methylation. a.u. = Arbitrary units.

The following figure supplement is available for figure 2:

**Figure supplement 1**. Differences in nucleosome occupancy between DNMT3b-expressing and non-expressing yeast strains.

From these analyses, it is evident that DNMT3b preferentially methylates non-nucleosomal DNA. We observe a 50% increase in the methylation of linker DNA compared to nucleosome bound DNA (*Figure 2C*). We also observe a slight 10 bp periodicity of methylated CpG (*Figure 2D*), another feature shown in higher eukaryotes that reflects the periodicity of the DNA helix (*Klug and Lutter, 1981*).

## Impact of DNA methylation on yeast nucleosome position and gene expression

We considered the possibility that introducing 5^meC would alter nucleosome distribution or gene expression in yeast. However, a comparison of DNMT3b-expressing and non-expressing strains showed no detectable change in nucleosome positioning by MNase treatment near the TSS, TTS (*Figure 2—figure supplement 1A,B* and *Supplementary file 3C*), or elsewhere in the genome.

RNA-seq analysis identified some differentially expressed genes (about 5% of the total number of genes, with an equal number of up- and down-regulated transcripts) between the strain expressing and non-expressing DNMT3b grown to stationary phase (*Figure 3* and *Supplementary file 1C* and *Supplementary file 4A*). The down-regulated genes showed enrichment for branched-chain aminoacid biosynthesis genes, while the up-regulated ones were enriched in ribosomal biogenesis

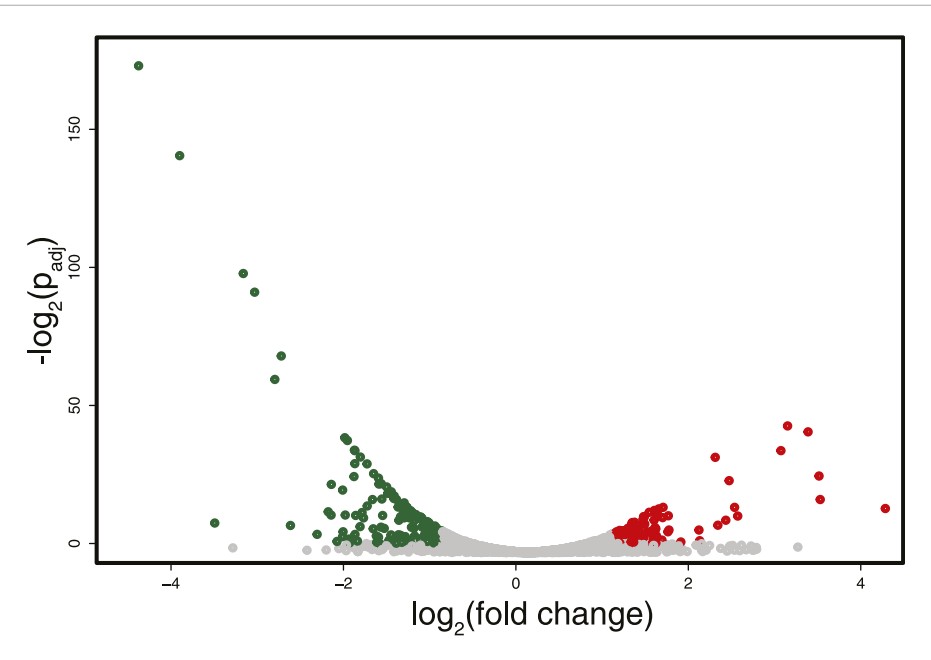

**Figure 3**. Differences in RNA expression between DNMT3b-expressing and non-expressing yeast strains. The expression difference in RNA expression between DNMT3b and EV strains is plotted on the x axis, and false discovery rate (FDR)-adjusted significance is plotted on the y-axis (–log2 scale). Upregulated and downregulated RNAs shown in red and green, respectively. Significantly expressed RNAs have a fold change bigger than two with a FDR smaller than 0.1.

The following figure supplements are available for figure 3:

**Figure supplement 1**. DNA Methylation in up- and down-regulated genes.

**Figure supplement 2**. DNA Methylation in ribosomal biogenesis genes.

genes (*Supplementary file 4B–F*). However, these changes are likely due to stress response pathways that are triggered by the overexpression of MmDNMT3b, rather than by the changes in DNA methylation itself. In support of this view, when the levels of CpG, CpHpG, and CpHpH methylation in the up- and down-regulated genes were compared, no significant difference was evident (*Figure 3—figure supplement 1*). Moreover, the methylation levels of the differentially transcribed genes were not different from that of other members of the same Gene Ontology (GO) term (*Figure 3—figure supplement 2*). Since DNA methylation machinery is not native in yeast, it is likely that proteins able to recognize and mediate 5$^{me}$C effects are also absent.

## DNMT3b activity is associated with specific histone tail modifications

We next sought to test whether the observed levels of 5$^{me}$C could be explained by the underlying distribution of specific histone tail modifications. To address this, we mapped the distribution of DNMT3b and of specific histone residue modifications via ChIP-seq in both the DNMT3b-expressing and wild type (wt) (non-expressing) strains (*Supplementary file 1D*).

We found that, as expected, DNMT3b co-localizes with methylated regions (*Figure 4A*). The distribution of DNMT3b is also consistent with the distribution of DNA methylation across the gene body (*Figure 4—figure supplement 1*). We also observed that DNMT3b and 5$^{me}$C are strongly anti-correlated with H3K4me3 and positively correlated with H3K36me3 (*Figure 4B* and *Figure 4—figure supplements 2–4*). By examining the distribution of histone marks across gene bodies, we found that H3K4me3 is concentrated at the promoter while H3K36me3 levels peak near the 3′ end of the gene (*Figure 4—figure supplement 1*). These observations suggest that the ADD and PWWP domains of DNMT3B play a role in targeting the activity of the enzyme. H3K4me1 shows a weak positive

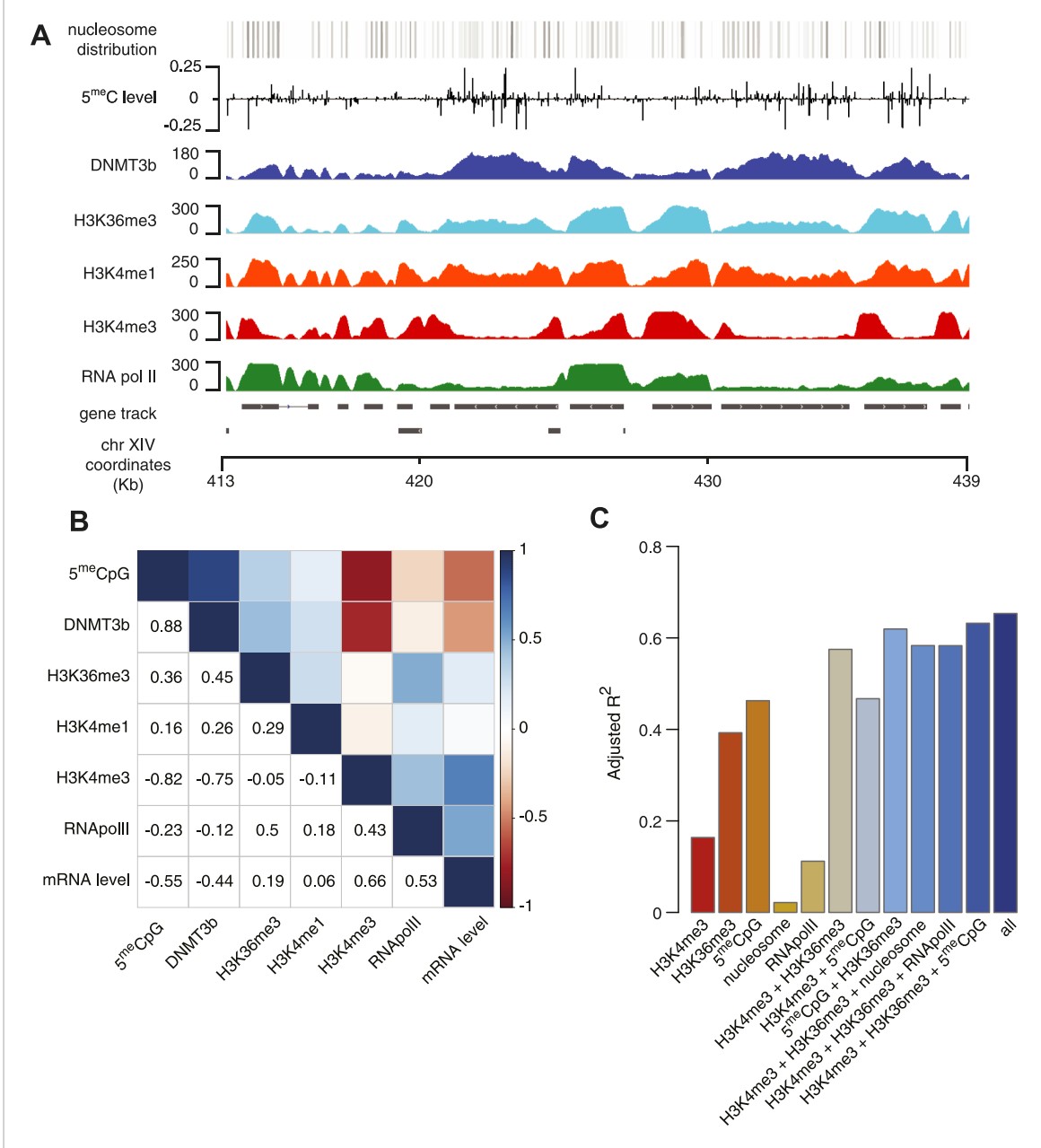

**Figure 4**. Correlation between histone marks and DNA methylation. (**A**) Genome-wide distribution of nucleosome, 5meC, DNMT3b, H3K36me3, H3K4me1, H3K4me3, and RNA polymerase II. (**B**) Spearman correlation coefficients between 5meC, histone marks, RNA polymerase II, DNMT3b and mRNA average levels for protein coding genes. (**C**) Prediction of DNMT3b levels using DNA methylation, H3K4 and H3K36 trimethylation, RNA polymerase II and nucleosome distribution as predictors. The y-axis shows the adjusted $R^2$ value between the predicted linear model and observed values.

The following figure supplements are available for figure 4:

**Figure supplement 1**. Metagene plot of ChIP sequencing in a DNMT3b-expressing strain.

**Figure supplement 2**. Relationship between transcription and 5meC or histone marks levels.

**Figure supplement 3**. Relationship between DNA methylation and histone marks levels.

*Figure 4. continued on next page*

*Figure 4. Continued*

**Figure supplement 4**. Relationship between H3K4me3 and 5meC or histone marks levels.

**Figure supplement 5**. 5meC levels prediction using chromatin marks.

correlation with both 5$^{me}$C levels and DNMT3b. This might be due to the specific distribution of H3K4me1 within the gene body, partially overlapping to the H3K36me3 modification (*Figure 4A* and *Figure 4—figure supplement 1*).

5$^{me}$C and DNMT3b distribution are also inversely correlated with gene transcription and Pol II abundance (*Figure 4B*, *Figure 4—figure supplements 2E–F, 3A*). Both H3K4me3 and H3K36me3 correlate positively with transcription (*Figure 4B*, *Figure 4—figure supplement 2C–D*). Since yeast genes are very small relative to mammalian genes, H3K4 methylation can spread well into the gene body (*Figure 4—figure supplement 2B–C*) and limit the deposition of 5$^{me}$C in highly transcribed genes. In support of this observation, we find that a higher level of H3K4me3 in the last third of the gene, is associated with a lower level of DNMT3b or 5$^{me}$C (*Figure 4—figure supplement 4*).

To determine whether the methylation of H3K4 and H3K36 is sufficient to explain the observed DNA methylation of our DNMT3b strains, we constructed a simple linear model of DNA methylation based on our ChIP-seq data. We used linear multivariate regression to model whether the distribution of one or a few histone marks, nucleosome positioning or RNA polymerase II occupancy could predict the levels of DNMT3b or 5$^{me}$C (*Figure 4C* and *Figure 4—figure supplement 5*). Strikingly, we found that H3K4me3 and H3K36me3 levels are sufficient to predict the distribution of both DNMT3b and 5$^{me}$C with very high accuracy. The prediction could only be slightly improved by using additional data, suggesting that H3K4me3 and H3K36me3 are the key factors in determining the targeting of DNA methylation (*Supplementary file 5*).

## Deletion of histone lysine methyltransferases affect DNA methylation distribution

To determine whether H3K36me3 has a direct role in the recruitment/activity of DNMT3b in vivo, we measured the DNA methylation distribution in three mutant strains: *set1Δ*, *set2Δ*, and *dot1Δ* (*Supplementary file 1E*). In yeast, Set1 is responsible for the methylation of H3K4, Set2 is the methyltransferase for H3K36, and Dot1 catalyzes the methylation of H3K79. We included the *dot1Δ* strain as a control, since we do not expect its activity to influence the binding of DNMT3b. If the modification of H3K36 plays a role in DNMT3b activity we would expect a reduction in DNA methylation levels in gene bodies, which are the primary H3K36me3 positive regions.

Due to an impact of the set mutations on global transcription, the levels of the induced DNMT3b and the resulting DNA methylation were lower in deletion strains than the wt. Nonetheless, the resulting 5$^{me}$C levels were still significantly higher than background levels found in the wt strains (*Figure 5A* and *Supplementary file 2B*). To account for the variations in global methylation levels we adopted two types of normalization: the first normalized by the total amount of DNA methylation in the sample and the second was based on the expression of DNMT3b measured via RT-qPCR (*Figure 5B* and *Figure 5—figure supplement 1*). Both strategies gave similar results (data not shown).

As expected, we see no significant differences in 5$^{me}$C distribution in *dot1Δ* strains compared to wt (*Figure 5B*). In contrast, in the *set1Δ* strain, we found that regions close to the TSS, with high H3K4me3 and low DNA methylation in a wt strain, contain methylation levels that are not significantly different from other regions outside of the gene (*Figure 5B,C*). This suggests that H3K4 methylation plays an active role in suppressing DNA methylation in the wt, and that this effect disappears in the *set1Δ* strain (*Figure 5D*).

In a *set2Δ* strain, 5$^{me}$C levels are reduced over gene bodies compared to wt strains (*Figure 5C*). Moreover, in this strain maximum levels of DNA methylation peak outside of the gene, where H3K36me3 is not present (*Figure 5B*). Thus, in this mutant strain DNA methylation is redistributed from gene bodies (H3K36me3-rich regions) to intergenic regions compared to the wt, suggesting that H3K36me3 is responsible for recruitment of DNMT3B (*Figure 5C,E*).

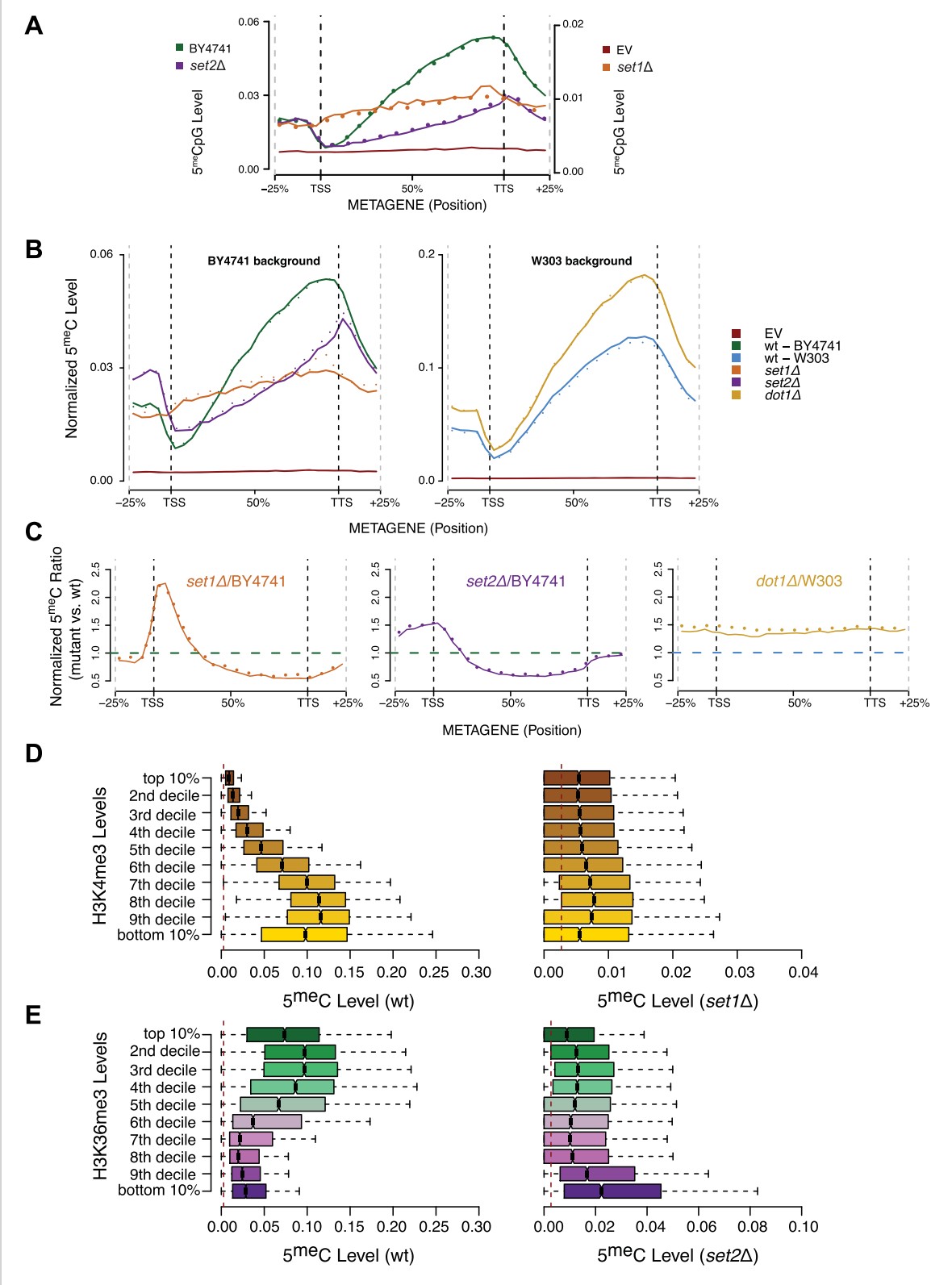

**Figure 5**. Effect of histone lysine methyltransferase deletions on the distribution of DNA methylation. (**A**) Metagene plot of CpG methylation in *set1Δ* and *set2Δ* cells expressing DNMT3b. Differently from *Figure 5B*, 5meC levels are not normalized. Replicates of the same strain are represented as dotted lines. Data from BY4741-derived strains. BY4741 = Wild type (wt); EV = Empty vector. (**B**) Metagene plot of CpG methylation in *set1Δ*, *set2Δ*, and *dot1Δ* cells expressing DNMT3b. *set1Δ*, *set2Δ* are in a BY4741 background, while *dot1Δ* is in a W303 background. 5meC levels are normalized by DNMT3b
*Figure 5. continued on next page*

## eLIFE Research article

Developmental biology and stem cells | Genomics and evolutionary biology

*Figure 5. Continued*

expression measured by RT-qPCR. Two replicates for each strain are shown (solid and dotted line). (**C**) Metagene plots of CpG methylation ratio between the mutant and its wt counterpart. Two replicates for each mutant strain are shown (solid and dotted line). Wt ratios (=1) are represented by the horizontal dashed line (green or blue). (**D**) Boxplots showing levels of DNA methylation in the wt (left) and *set1Δ* strain (right) of 200-bp genome bins sorted into deciles by H3K4me3 level. (**E**) Boxplots showing levels of DNA methylation in the wt (left) and *set2Δ* strain (right) of 200-bp genome bins sorted into deciles by H3K36me3 level. The dashed red line represents background levels of DNA methylation due to incomplete bisulfite conversion (>99.7%).

The following figure supplement is available for figure 5:

**Figure supplement 1**. DNMT3b transcript levels in different yeast strains.

## Correspondence between H3K36me3 and early DNA methylation in mammalian cells

To extend our findings in yeast, we sought evidence to determine whether H3K36me3 also promotes de novo DNA methylation in mammals. The mouse germline is an excellent model for such studies. The mouse germline is specified from the epiblast at E7.25 and then progressively loses DNA methylation through subsequent rounds of cell division. By E13.5, almost all DNA methylation has been lost (*Popp et al., 2010*; *Seisenberger et al., 2012*). In male germ cells, cell division halts, and the de novo methyltransferases and their co-factor DNMT3L are expressed between E13.5 and birth, when the genome undergoes global de novo DNA methylation (*Seisenberger et al., 2012*; *Kobayashi et al., 2013*). Thus in this setting, DNA methyltransferases are introduced into hypomethylated cells, and are therefore an ideal model to study the targeting of de novo DNA methylation.

We mapped DNA methylation in the male germline at E16.5, P2.5, and P10.5 (*Supplementary file 1F*) (*Pastor et al., 2014*), and obtained E13.5 DNA methylation data from published sources (*Seisenberger et al., 2012*). Consistent with previous observations about the timing of de novo DNA methylation in the developing mouse germline, global CpG methylation rises from 7% at E13.5 to 55% at E16.5 and reaches at 75% by P2.5 (*Figure 6A*). Previous studies have shown that the entire male germline genome is methylated by default, except for regions of H3K4 methylation such as TSSs which antagonize de novo DNA methylation (*Singh et al., 2013*). However, charting the progression of DNA methylation over time, it is apparent that there exist 'early methylating' regions that reach their final methylation state by E16.5 and 'late methylating' regions that undergo substantial DNA methylation between E16.5 and P2.5. We observed that heavily transcribed regions of chromosomes showed much higher DNA methylation at E16.5 than less transcribed regions (*Figure 6B*). Furthermore, while the TSS of active genes was unmethylated, gene bodies of actively transcribed genes were typically early-methylators (*Figure 6B,D*). Thus, transcriptional initiation correlates negatively with de novo DNA methylation while transcriptional elongation correlates positively with de novo methylation.

In light of the data from yeast, we considered that the trends noted above could be caused by the underlying chromatin environment, with H3K4me3 antagonizing and H3K36me3 promoting de novo DNA methylation. Since transcriptional elongation causes H3K36me3 deposition, we asked whether the association of transcriptional read-through with DNA methylation could explain the observed phenomenon. To test this hypothesis, we analyzed published H3K4me3 ChIP-seq data (*Lesch et al., 2013*) and performed H3K36me3 ChIP-seq on sorted germ cells of pooled E13.5 testis (*Supplementary file 1G*). H3K4me3 at E13.5 correlates with low DNA methylation at all subsequent time points (*Figure 6D,E*). Genes with high H3K36me3 levels at E13.5 showed significantly elevated gene-body DNA methylation at E16.5, consistent with H3K36me3 accelerating DNA methylation (*Figure 6B,C,D,F*). This trend was still apparent at P2.5 (*Figure 6F*). Thus, H3K36me3 appears to direct DNA methylation in mammalian cells.

## Discussion

Our study aimed to identify chromatin features that affect the activity of mammalian de novo DNMTs in the establishment of DNA methylation. The expression of the murine DNMT3b in a host with no detectable levels of 5^meC led to the methylation of CpG dinucleotides at different levels depending on the specific chromatin context. The presence of the H3K4me3 mark inhibits the activity of DNMT3b, while H3K36me3 promotes DNA methylation. This suggests that the activity of DNMT3B is guided by

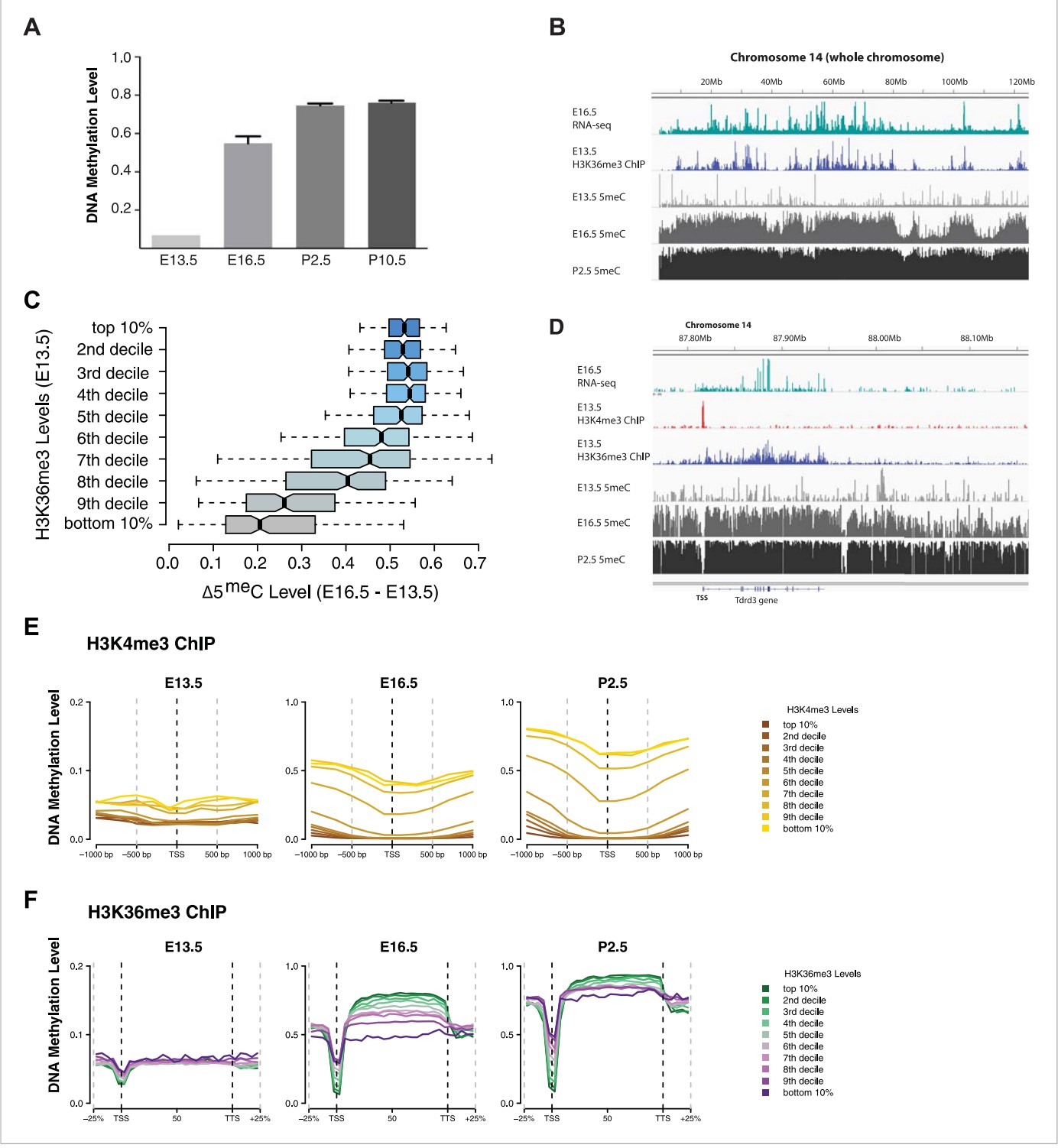

**Figure 6**. H3K4me3 and H3K36me3 distribution predicts de novo DNA methylation pattern in male germline. (**A**) Genome-wide CG methylation levels during murine development as measured by bisulfite sequencing. (**B**) RNA-seq, and ChIP read abundance and relative DNA methylation levels are plotted across chromosome 14. Note the correspondence between RNA-seq and H3K36me3 ChIP levels and rapid DNA methylation between E13.5 and E16.5. (**C**) Boxplots showing the difference of DNA methylation levels between E13.5 and E16.5 of 1 Mb genome bins sorted into deciles by H3K36me3 level. (**D**) RNA-seq and ChIP read abundance and DNA methylation levels are plotted relative to transcriptionally active genes. The gene promoters contain high H3K4me3 and are not methylated, while the gene bodies contain high H3K36me3 and are methylated rapidly. (**E**) Metaplots showing DNA methylation level ±1000 bp relative to the TSS of genes sorted into deciles by H3K4me3 level. (**F**) Metagene plots showing DNA methylation across gene bodies sorted into deciles by H3K36me3 level.

the interactions of the ADD and PWWP domains with histone tails. It has been recently shown (*Baubec et al., 2015*) that in embryonic stem cells the PWWP domain is responsible for the targeting of DNMT3b to regions enriched for the H3K36me3. Similarly to our finding in yeast, the reintroduction of DNMT3b into methylation deficient DNMT1/DNMT3A/DNMT3B triple KO (TKO) ES cells partially restores 5$^{me}$C levels. Methylation levels are higher at H3K36me3 sites, a trend eliminated by the ablation of the H3K36me3 methyltransferase Setd2 (*Baubec et al., 2015*). Our findings are in agreement with the Baubec et al. observations, both in a system where other factors guiding DNA methylation are absent (yeast), and during a period of biologically important de novo DNA methylation (germ cells).

In our yeast system, we detected an anti-correlation between transcript levels and DNA methylation, while we found a positive correlation in germ cells as was shown in ES cells (*Baubec et al., 2015*). According to our findings, the levels of DNA methylation are guided by the presence of two transcription-dependent marks: H3K4 and H3K36 methylation. The discrepancy between the findings in yeast and germ cells can be explained by the difference in the length of their genes. Yeast genes are relatively small compared to genes in higher eukaryotes so, H3K4 methylation can spread within the body of the gene, thus preventing the binding of the DNMT3-ADD domain to the N-terminus of histone H3 and reducing its activity. In contrast, in mammals, H3K4me is localized to the start of the gene, and does not spread significantly within the gene body. Hence, highly transcribed genes in mammals show a strong enrichment of H3K4me3 around the TSS and H3K36me3 into the gene-body, shaping their intragenic DNA methylation distribution.

The observation that transcriptional elongation is linked to DNA methylation has been noted in many contexts in addition to male germ cells. In mature oocytes, which have intermediate global levels of CpG methylation (~50%), similar to male E16.5 PGCs, actively transcribed gene bodies have far higher levels of DNA methylation than less transcribed genes and intergenic regions (*Smallwood et al., 2011*; *Kobayashi et al., 2012*). Also, in oocytes, intragenic CpG islands show far higher DNA methylation than other CpG islands (*Smallwood et al., 2011*). Transcriptional read-through is a common feature of maternally imprinted loci (*Weaver and Bartolomei, 2014*) and ablation of an upstream promoter prevents proper methylation of the imprinted *Gnas* locus (*Chotalia et al., 2009*). In mammalian soma, inactive X-chromosome shows higher promoter methylation, consistent with its silent state, but markedly lower intragenic methylation (*Hellman and Chess, 2007*). Transcriptional elongation is also correlated with DNA methylation in tumor cells (*Jin et al., 2012*). It has been suggested that transcriptional read-through could 'open' chromatin for DNMTs, or that heterochromatin is physically inaccessible to DNMTs. We suggest however that direct recruitment of DNMTs by H3K36me3 is the most likely mechanism for the correlation between transcriptional read-through and DNA methylation.

H3K36me3 functions both to suppress intragenic transcriptional initiation through recruitment of histone deacetylases, and to promote DNA methylation. These marks likely cooperate to induce lasting silencing of transcriptional initiation at target loci (*Figure 7*, *Figure 7—figure supplement 1*). Intragenic TSSs originating at transposons have the potential to generate truncated or transposon/gene hybrid transcripts that could be deleterious to cell survival. H3K36me3 and DNA methylation could cooperate to silence these transposons in the germline and other periods of de novo methylation, and to maintain silencing through development. Moreover, where multiple TSSs exist for a gene, as in many imprinted loci, H3K36me3-mediated DNA methylation may serve to ensure the dominance of one promoter in a given cell type.

A number of H3K36 methyltransferases exist in mammals but only one, SETD2, can catalyze the conversion of H3K36me2 to me3 (*Wagner and Carpenter, 2012*). *Setd2*$^{-/-}$ mice exhibit profound vascular defects and die at E10.5–E11.5 (*Hu et al., 2010*), while *Setd2*$^{-/-}$ are defective for differentiation toward endoderm (*Zhang et al., 2014*). *Setd2* is also a tumor suppressor mutated frequently in leukemia (*Zhu et al., 2014*). It will be important to determine how loss of *Setd2* affects the distribution of DNA methylation in the germline and soma, and whether loss of *Setd2* contributes to aberrant methylation in cancer.

More broadly, targeting of DNMT enzymes by association with H3K36me3 could explain methylation distribution across plants and animals. All catalytically active DNMT3-family methyltransferases in animals contain PWWP domains, and accordingly, gene body DNA methylation is observed in all animals that have retained DNMT3 enzymes. Preferential methylation of gene bodies over intergenic regions is observed for invertebrates such as honey bees (*Apis mellifera*), sea squirts (*Ciona intestinalis*), sea anemones (*Nematostella vectensis*) (*Zemach and Grafi, 2003*; *Feng et al., 2010*). While the relationship between relative gene expression and gene-body methylation varies

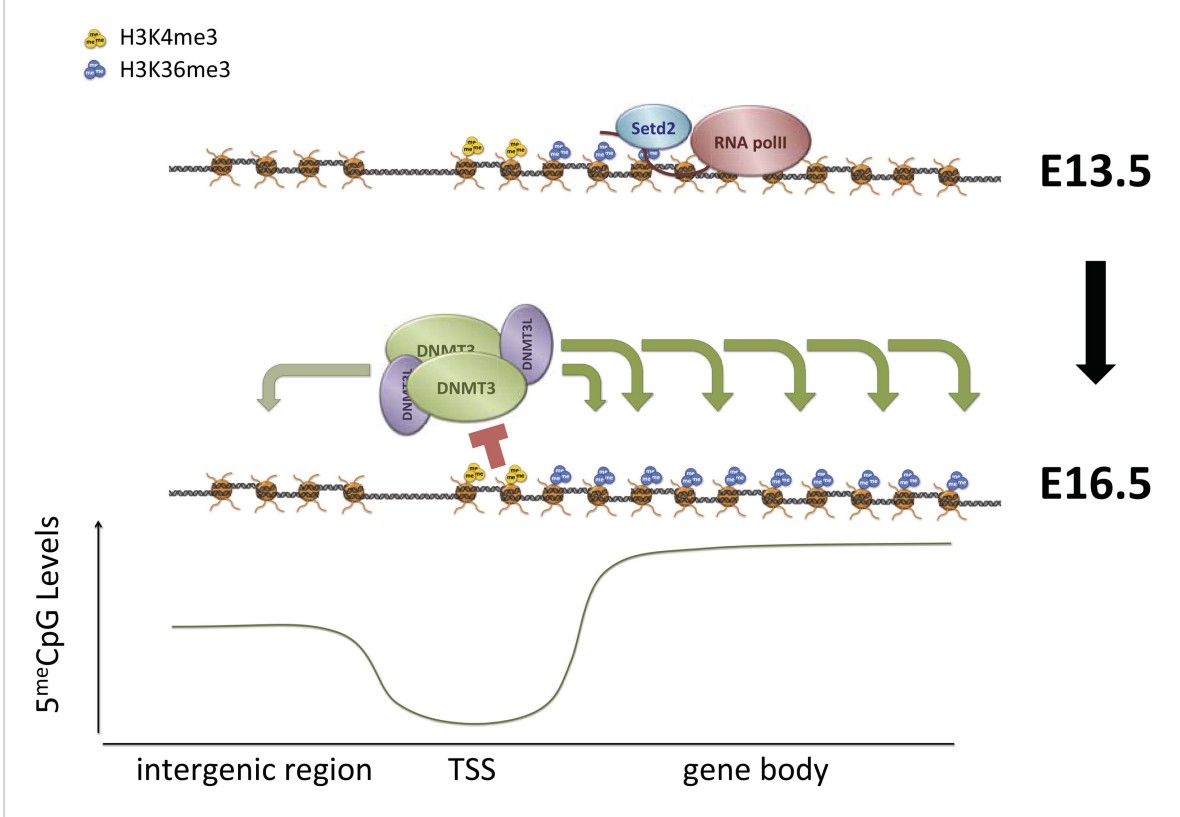

**Figure 7**. Proposed model for de novo DNA methylation establishment. Model proposed for the targeting of DNMT3 during events of de novo 5meC establishment after genome-wide erasure of DNA methylation. Our model suggests that the presence of transcription-dependent histone modifications, such as H3K4me3 and H3K36me3, determines the activity of DNMT3b in vivo.

The following figure supplement is available for figure 7:

**Figure supplement 1**. Factors affecting DNA methylation deposition.

across these species, there is a strong correlation between gene-body H3K36me3 in *Drosophila melanogaster* genes and DNA methylation of homologous gene bodies in other invertebrates (*Nanty et al., 2011*). DNA methylation is also associated with gene bodies in zebrafish (*Danio rerio*) (*Zemach and Grafi, 2003*) and in mammalian contexts as discussed above. Finally, some chlorophyte algae have a 'chlorophyte-type cytosine methylase', which evolved independently of DNMT3-family methyltransferases, which is fused to two PWWP domains (*Iyer et al., 2011*). Thus, H3K36me3 could be relevant to DNA methylation targeting throughout the plant and animal kingdoms.

## Materials and methods

### Experimental methods

#### Yeast strains, plasmids, media and culture Conditions

All the plasmids, primers, and strains used in this study are listed in *Supplementary file 6A,B*. Murine DNMT3b isoform 1 was amplified from the plasmid pCR-Blunt II-TOPO and subcloned into pYES2 (Life Technologies, Carlsbad, CA) using *Hind*III and *Bam*HI. All the plasmids were introduced in yeast using the standard Lithium Acetate Procedure (*Gietz and Schiestl, 2007*). Mutant yeast strains *set1Δ* and *set2Δ* were kindly donated by the Kurdistani Lab (UCLA), while the *dot1Δ* strain (W303 background) was prepared via PCR-mediated gene disruption (*Wach, 1996*) using primers listed in *Supplementary file 6C*.

All the yeast strains were grown at 30°C in SC + Galactose 2% without uracil (Sunrise Science, San Diego, CA, cat 1652 and 1485-100) overnight. The next morning, cells were diluted to 0.3 $OD_{600}$/ml and grown to mid-log phase (0.8–1 $OD_{600}$/ml) or to stationary phase (5–6 $OD_{600}$/ml or for 24–30 hr).

## Yeast WGBS libraries preparation

DNA was collected from yeast cells according to Hoffman (2001) with minor changes. Briefly, about 5 $OD_{600}$ of yeast cells were disrupted by vortexing for 6 min in a Disruptor Genie (Scientific Industries, Inc., Bohemia, NY) in the presence of an equal volume of breaking buffer, acid-washed glass beads and phenol:chloroform (1:1). After the addition of TE buffer, aqueous phase is transferred into a new tube and precipitated with ethanol. The nucleic acid pellet is resuspended in TE buffer and treated for 1 hr at 37°C with RNaseA, followed by incubation for 1 hr with 2 mg/ml proteinase K in the presence of 1% SDS at 60°C. The resulting solution is treated twice with phenol:chloroform (1:1), once with chloroform and ethanol precipitated. The DNA pellet is resuspended in EB buffer (Qiagen, Valencia, CA). Between 500 and 1000 ng of extracted yeast DNA is added to 2 ng of λ unmethylated DNA (Promega, Madison, WI, D1521) and the mixture is sonicated with a Covaris S-2 to obtain fragments in the 200–300 bp range (Total time: 6 min; Duty cycle: 10%; Intensity: 5; Cycles/Burst: 200; Mode: Frequency sweeping). The reagents used in the library preparation are from the Illumina TruSeq DNA Sample Prep kit v2 (Illumina, San Diego, CA). End-Repair, purification and dA-tailing steps are performed according to manufacturer's instructions. Ligation is performed according to the protocol except that 1 μl of Illumina TruSeq Adapters is used in the final reaction. The ligation reaction is purified using 1.2 vol of AMPure XP beads (Beckman Coulter Inc. Indianapolis, IN,) and DNA fragments with a 170–350 bp range are enriched using 0.7 and 0.3 vol of AMPure XP beads in the first and second size-selection step, respectively. Samples are treated with bisulfite (EpiTect kit, QIAGEN) according to manufacturer's protocol, except that two consecutive rounds of conversion are performed, for a total of 10 hr of incubation. Half of the converted DNA is amplified using MyTaq Mix (Bioline, Taunton, MA,) and Illumina TruSeq PCR Primer Cocktail according to the following protocol: initial denaturation at 98°C for 30 s; 12 cycles of 98°C for 15 s, 60°C for 30 s, 72°C for 30 s; final extension at 72°C for 5 min. The final product is purified using AMPure XP beads before being submitted for sequencing. Libraries were sequenced with an Illumina HiSeq 2000 system using 50 bp or 100 bp single-end reads.

## Yeast MNase-seq libraries preparation

Nucleosome mapping has been performed according to Rando (2010) with minor modifications. Stationary phase yeast culture (≈6 OD/ml) is cross-linked with 1% formaldehyde for 20 min with occasional rotation at room temperature. The reaction is quenched with glycine for 5 min at room temperature. About 60 OD of yeast cells are washed twice with PBS buffer and then resuspended in 2 ml of Z buffer (1 M sorbitol, 50 mM Tris-HCl pH 7.4 with freshly added 10 mM β-mercaptoethanol) containing 3.6 mg of Zymolyase-20T (from Arthrobacter luteus, AMS Biotechnology, Cambridge, MA,) and incubated at 37°C in agitation. After 45 min the same amount of Zymolyase is added and each sample which is incubated for an additional 45 min at 37°C in agitation. Spheroplasts are then pelleted by centrifugation for 5 min at 4°C at 1500 g. The pellet is washed once with NP-buffer, then resuspended in 1.6 ml of NP-buffer and divided in three tubes. An increasing amount of MNase (Sigma, N3755, St. Louis, MO,) is added to each tube: 0.25 U, 0.5 U, and 1 U. After incubation for 20 min at 37°C, each reaction is stopped by the addition of SDS and EDTA to a final concentration of 1% and 10 mM, respectively. The reaction is then treated with 0.2 mg/mg of proteinase K (NEB, Ipswich, MA) at 65°C overnight. The sample is then purified with two rounds of phenol:chloroform (1:1) and the aqueous solution precipitated. The resuspended DNA pellet is treated for 1 hr with RNase A at 37°C. For the naked DNA digestion, 200 ng of extracted DNA is incubated at 37°C with 0.01 U of MNase. After 7 min the reaction is stopped as described before. Both naked and RNaseA-treated nucleosomal DNAs are then purified using 1.8 vol of AMPure XP beads and the libraries prepared using NEBNext DNA Library Prep Master Mix Set for Illumina (NEB, E6040S) with few modifications of the manufacturer's protocol. Only the digestion pattern obtained with 0.5 U of MNase was used for the library preparation. The DNA is end-repaired (in half of the suggested volume), dA-tailed, and 1 μl of Illumina TruSeq Adapters is added to a 40 μl ligation reaction. Purification after each step is performed using AMPure XP beads according to the protocol. The size selection step is carried out with 0.8× of AMPure beads in the first step and 0.2× of AMPure XP beads in the second step. Half of the DNA is amplified using Illumina PCR Master Mix and Illumina TruSeq PCR Primer Cocktail (TruSeq DNA Sample Prep kit v2) with the following protocol: initial denaturation at 98°C for 30 s; 12 cycles of 98°C for 15 s, 60°C for 30 s, 72°C for 30 s; final extension at 72°C for 5 min. The final product is purified using AMPure XP beads before being submitted for sequencing. Libraries were sequenced with an Illumina HiSeq 2000 system using 50 bp single-end reads.

## Yeast RNA-seq libraries preparation

RNA was collected from 5 OD of yeast cells (*Collart and Oliviero, 2001*). Approximately 500–1000 ng of extracted yeast RNA are used to prepare RNA-seq libraries using Illumina TruSeq mRNA Library Prep Kit v2 according to manufacturer's instructions. Libraries were sequenced with an Illumina HiSeq 2000 system using 50 bp single-end reads.

## RT-qPCR

Quantitative RT-PCR was used to determine the relative expression of DNMT3b in wild-type and mutant yeast strains. Briefly, 1 µg of total RNA was subject to polyA enrichment using TruSeq oligo-dT magnetic beads (part # 15026778, Illumina) and reverse transcribed using SuperScript III (cat # 18080-044, Life Technologies) according to manufacturer's instruction. An equal amount of cDNA was used for each qPCR reaction, using primers listed in *Supplementary file 6C*. Murine DNMT3b expression levels were normalized to TDH1 levels and the relative expression between the wild-type and each mutant was calculated using the ΔΔCt method (*Schmittgen and Livak, 2008*).

## Yeast ChIP-seq libraries preparation

Chromatin immunoprecipitation experiments were conducted according to *Kitada et al. (2012)*, with minor modifications. Briefly, 50 OD of yeast cells are crosslinked using 1% formaldehyde for 15 min at room temperature and quenched with glycine 125 mM for 5 min at room temperature. After two washes with ice-cold PBS, the cells are resuspended in yeast lysis buffer (with 140 mM NaCl for DNMT3b and RNApolII or 500 mM NaCl for histone post-translational modifications) and the same volume of acid-washed glass beads. We disrupted the cells by vortexing for 5 min in a Disruptor Genie at 4°C and incubating in iced-water for 2 min. We repeated the cycle for an additional 5 times. We collected the lysate by centrifugation after creating a hole on the bottom of the tube with a 25-G needle. We transferred a fraction of the lysate into a microTube (AFA filter—Covaris, Woburn, MA) and proceeded with the sonication using the Covaris S2 system according to the following parameters: 14 cycles of 30 s ON, 30 s OFF; Duty cycle = 5%; Intensity = 5%; Cycles/Burst = 200. The sonicated lysate is clarified via centrifugation and 50 µl of the supernatant is incubated overnight at 4°C with a specific antibody (*Supplementary file 6D*). 10 µl of the clarified lysate is used as input control. The next day, immunoprecipitations are incubated 2 hr at 4°C with Protein A Dynabeads (Life Technologies). Each wash is performed twice in the following order: low-salt buffer (50 mM HEPES pH 7.5, SDS 0.1%, 1% Triton X-100, 0.1% Deoxycholate, 1 mM EDTA, 140 mM NaCl), high salt buffer (50 mM HEPES pH 7.5, SDS 0.1%, 1% Triton X-100, 0.1% Deoxycholate, 1 mM EDTA, 500 mM NaCl), LiCl buffer (10 mM Tris-HCl pH 8, 250 mM LiCl, 5 mM EDTA, 1% Triton-X, 0.5% NP-40), TE buffer (100 mM Tris-HCl pH 8, 10 mM EDTA). Elution is performed at 65°C with TE/SDS buffer (100 mM Tris-HCl pH 8, 10 mM EDTA, 1% SDS). Tubes containing the eluted immunoprecipitations and input controls (additioned of TE/SDS buffer) are incubated overnight at 65°C to reverse the cross-links. RNase treatment is performed at 37°C for 1 hr, followed by a proteinase K treatment for 1 hr at 60°C. Each reaction is then purified using 1.8 vol of AMPure XP beads according to manufacturer's instructions. Libraries were prepared with Ovation Ultralow DR kit (Nugen Technologies, San Carlos, CA) starting from 1 ng of purified DNA according to the protocol. Libraries were sequenced with an Illumina HiSeq 2000 system using 50 bp single-end reads.

## Mice

Mice homozygous for a characterized Oct4-IRES-GFP allele (*Wernig et al., 2007*) were used for murine H3K36me3 ChIP. Embryonic male germ cells express the GFP marker and can be sorted efficiently (*Vincent et al., 2011*; *Pastor et al., 2014*).

## Bisulfite sequencing and RNA-seq data (mouse)

Whole genome bisulfite sequencing data from sorted E16.5, P2.5, and P10.5 germ cells was generated as part of a parallel project studying the transposon silencer *Morc1* (*Pastor et al., 2014*), with the data from the phenotypically normal *Morc1*[+/−] controls from that study serving as methylomes in this study. Briefly, germ cells from between three to five male mice at each time point were harvested and libraries generated, and reads from these libraries were pooled. E13.5 bisulfite sequencing data were taken from replicate two of (*Seisenberger et al., 2012*). Genome-wide bisulfite sequencing average coverage was 5.36 (E13.5), 14.57 (E16.5), and 8.52 (P2.5).

RNA-seq data from two E16.5 *Morc1*[+/−] controls from (*Pastor et al., 2014*) were also used in this study.

## Mouse germ cells purification for ChIP

Collection of embryonic testes was performed following institutional approval for appropriate care and use of laboratory animals, according to published protocols (*Pastor et al., 2014*). Pregnant females were euthanized using CO$_2$ and the embryos removed from the womb and stored on a 10 cm dish filled with chilled 1× PBS. Testicles were removed from the embryos, placed in an individual 15 ml falcon tube with 3 ml of 0.25% Trypsin with 3 µl of DNAse I 1 U/1 µl (Life Technologies). Testes were incubated for 15 min at 37°C. After incubation the cells were agitated into suspension gently by pipetting. The trypsin was then quenched using 5 ml DMEM/10% FBS (Life Technologies). The cells were centrifuged at 278 g for 5 min and resuspended in 500 µl FACS buffer (1× PBS 1% BSA). 7-AAD was added at a 1:50 dilution (BD Biosciences, San Jose, CA) and the cells strained through BD FACS tubes (Corning, Union City, CA) before analysis. GFP positive cells were sorted for ChIP.

## Mouse ChIP-seq

The ChIP-seq protocol was adapted from published sources (*Ng et al., 2013*; *Pastor et al., 2014*). FACS sorted cells from four male, E13.5 embryos were diluted to 292 µl with room temperature 1× PBS. 8.11 µl 37% Formaldehyde (Sigma) was added and the sample was incubated 10 min at room temperature with rocking. 48.8 µl of 1 M glycine was then added to yield a final concentration of 0.14 M and the samples were quenched 30 min with rocking. Cells were then spun 425 g for 10 min at RT. The cell pellet was flash frozen.

After thawing, the cells were resuspended in 300 µl Lysis buffer (50 mM Tris-Cl pH 8.0, 20 mM EDTA pH 8.0, 0.1% SDS, 1× Complete Protease Inhibitor [Roche]) and incubated on ice 10 min. Samples were then sonicated by Covaris S2 (Intensity 5, cycles/burst = 200, duty cycle = 5%, 10 × 30 s on 30 s off sonication). Samples were spun 14000 g 10 min to remove insoluble material. The soluble sample was diluted to 600 µl with dilution buffer (16.7 mM Tris pH 8, 0.01% SDS, 1.1% Triton X-100, 1.2 mM EDTA, 167 mM NaCl) and 10% of material was saved as input. Sample was precleared with 30 µl Protein A Dynabeads (Life Technologies) and preincubated 1 hr. The cleared material was incubated with 1 µl L anti-H3K36me3 antibody (Abcam Ab9050) overnight.

The samples were incubated with 30 µl Protein A Dynabeads and the precipitated material was recovered with a magnet. The beads were washed 2 × 4 min with Buffer A (50 mM HEPES pH 7.9, 1% Triton X-100, 0.1% Deoxycholate, 1 mM EDTA, 140 mM NaCl), 2 × 4 min with Buffer B (50 mM HEPES pH 7.9, 0.1% SDS, 1% Triton X-100, 0.1% Deoxycholate, 1 mM EDTA, 500 mM NaCl) and 2 × 4 min with 10 mM Tris/1 mM EDTA. Bound material was eluted with 100 µl Elution buffer (50 mM Tris pH 8.0, 1 mM EDTA, 1% SDS) at 65°C for 10 min and then eluted a second time with 150 µl elution buffer.

The input samples were thawed and diluted with 200 µl buffer. Crosslinking of ChIP and input samples was reversed by incubating 16 hr at 65°C. Samples were cooled and treated with 1.5 µl of 10 mg/ml RNaseA (PureLink RNAse A, Invitrogen #12091-021) for 30 min at 37°C. 100 µg of Proteinase K was then added and the samples treated for 2 hr at 56°C. The samples were then purified using a Qiagen MinElute kit.

Samples were amplified by a SeqPlex DNA Amplification kit (Sigma) and then converted to libraries using an Ovation Rapid Library kit.

# Data processing and analysis

## Bisulfite sequencing

Reads from bisulfite-treated yeast and mouse genomic DNA (*Seisenberger et al., 2012*; *Pastor et al., 2014*) were aligned using BS-Seeker2 v2.0.3 (*Guo et al., 2013*) against the sacCer3 and mm9 genome assemblies, respectively. Up to four mismatches were allowed and bowtie (v0.12.8) was specified as the aligner. Methylation was called using default parameters of BS-Seeker2.

## MNase-sequencing

Reads from both naked and nucleosomal DNA sequencing were aligned using bowtie v0.12.8 (*Langmead et al., 2009*) against the sacCer3 genome assembly, allowing up to two mismatches. Nucleosome calling was performed using DANPOS v2.1.3 (*Chen et al., 2013*) subtracting the naked DNA-derived reads from the nucleosomal reads and using the '-k1 -e1' parameters (*Supplementary files 1B, 4*).

## RNA sequencing

RNAseq reads from mouse germ cells (*Pastor et al., 2014*) and yeast were aligned against the mm9 and sacCer3 genome assemblies using STAR v2.3.1 (*Dobin et al., 2013*) with the following parameters: –outFilterMismatchNoverLmax 0.04 –outFilterMultimapNmax 1.

Differential expression was performed using the DEseq package (*Anders and Huber, 2010*) in R-Bioconductor. Differentially expressed genes are defined as having more than twofold difference in the level of the corresponding RNA and a false discovery rate (p-adj) smaller than 0.1.

GO term enrichment for upregulated and dowregulated genes in the DNMT3b-expressing compared to the EV was performed using the Gene Ontology Term Finder tool on the Sccharomyces Genome Database website (http://www.yeastgenome.org/cgi-bin/GO/goTermFinder.pl). RPKM values were calculated using rpkmforgenes.py (available at http://sandberg.cmb.ki.se/media/data/rnaseq/rpkmforgenes.py) specifying the following options: -fulltranscript -nocollapse -rmnameoverlap –allmapnorm (*Supplementary file 4*).

### ChIP sequencing
Reads from yeast and mouse (this study and [*Lesch et al., 2013*]) were first mapped against the yeast (sacCer3) and mouse (mm9) genome, respectively, using bowtie v0.12.8 (*Langmead et al., 2009*), then aligned reads were processed according to *Ferrari et al. (2012)*.

### Linear model of methylation
The yeast genome was divided in 200-bp bins and log-transformed average levels of each feature calculated for each bin. The model was built using simple linear regression lm() function in R and the resulting prediction correlated (Pearson) with the observed values for both $5^{me}C$ levels and DNMT3b occupancy.

### Data access
Data can be accessed at GEO (Gene Expression Omnibus) under the accession GSE6691.

## Acknowledgements

We are grateful to Dr Maria Vogelauer for the helpful discussion and insights on the project. Yeast mutant strains were kindly provided by the Kurdistani Lab (UCLA). We also thank the Broad Stem Cell Research Center High-Throughput Sequencing and Flow Cytometry Cores.

## Additional information

### Funding

| Funder | Grant reference | Author |
| --- | --- | --- |
| National Institutes of Health (NIH) | R01 GM095656-01A1 | Matteo Pellegrini |
| National Institutes of Health (NIH) | GM60398 | Steven E Jacobsen |
| National Institute of Child Health and Human Development (NICHD) | R01HD058047 | Amander T Clark |
| Howard Hughes Medical Institute (HHMI) | | Steven E Jacobsen |
| Jane Coffin Childs Memorial Fund for Medical Research | | William A Pastor |
| Whitcome Fellowship | | Marco Morselli |

The funders had no role in study design, data collection and interpretation, or the decision to submit the work for publication.

### Author contributions
MM, WAP, Conception and design, Acquisition of data, Analysis and interpretation of data, Drafting or revising the article; BM, Conception and design, Acquisition of data, Drafting or revising the article; KN, LR, Acquisition of data, Drafting or revising the article; RF, KF, GB, Analysis and interpretation of data, Drafting or revising the article; ATC, SEJ, MP, Conception and design, Drafting or revising the article; SO, Drafting or revising the article, Contributed unpublished essential data or reagents

### Ethics
Animal experimentation: All animal experimentation was conducted with the highest ethical standards in accordance with UCLA policy and procedures (DHHS OLAW A3196-01, AAALAC

#000408 and protocol # 2008-070), and applicable provisions of the USDA Animal Welfare Act Regulations, the Public Health Service Policy on Humane Care and Use of Laboratory Animals, and the Guide for the Care and Use of Laboratory Animals.

# Additional files

## Supplementary files

• Supplementary file 1. (**A**) Yeast Whole Genome Bisulfite Sequencing Data. (**B**) Yeast MNase Sequencing Stats. (**C**) Yeast mRNA Sequencing Stats. (**D**) Yeast ChIP Sequencing Stats. (**E**) Yeast Whole Genome Bisulfite Sequencing Data for mutant strains. (**F**) Yeast Whole Genome Bisulfite Sequencing in mouse. (**G**) ChIP Sequencing Stats in mouse.

• Supplementary file 2. (**A**) Yeast dinucleotide context methylation. (**B**) Yeast mutant strains dinucleotide context methylation. (**C**) Mouse germ cells dinucleotide context methylation.

• Supplementary file 3. (**A**) Nucleosome called in a DNMT3b-expressing strain. (**B**) Nucleosome called in a non DNMT3b-expressing strain (EV). (**C**) Differential nucleosomes between DNMT3b-expressing and non-expressing strain.

• Supplementary file 4. (**A**) Yeast mRNA differential expression using Deseq. (**B**) Upregulated genes in DNMT3b-expressing strain vs EV. (**C**) Downregulated genes in DNMT3b-expressing strain vs EV. (**D**) Gene Ontology (GO) term analysis for upregulated genes in DNMT3b-expressing strain vs EV. (**E**) GO term analysis for downregulated genes in DNMT3b-expressing strain vs EV. (**F**) RPKM values of yeast verified ORF in DNMT3b-expressing strain.

• Supplementary file 5. Correlation coefficients of DNMT3b occupancy and 5$^{me}$C levels predictions.

• Supplementary file 6. (**A**) Plasmids used in this study. (**B**) Yeast strains used in this study. (**C**) Oligonucleotides used in this study. (**D**) Antibodies used in this study.

## Major datasets

The following dataset was generated:

| Author(s) | Year | Dataset title | Dataset ID and/or URL | Database, license, and accessibility information |
|---|---|---|---|---|
| Morselli M, et al., | 2015 | In vivo targeting of de novo DNA methylation by histone modifications in yeast and mouse | http://www.ncbi.nlm.nih.gov/geo/query/acc.cgi?token=anmpigiuppmnfeh&acc=GSE66911 | Publicly available at the NCBI Gene Expression Omnibus (GSE6691). |

The following previously published datasets were used:

| Author(s) | Year | Dataset title | Dataset ID and/or URL | Database, license, and accessibility information |
|---|---|---|---|---|
| Seisenberger S, et al., | 2012 | The dynamics of genome-wide DNA methylation reprogramming in mouse primordial germ cells | http://www.ebi.ac.uk/ena/data/view/ERP001953 | Publicly available at the EBI European Nucleotide Archive (ERP001953). |
| Pastor W, et al., | 2014 | MORC1 represses transposable elements in the mouse male germline | http://www.ncbi.nlm.nih.gov/geo/query/acc.cgi?acc=GSE63048 | Publicly available at the NCBI Gene Expression Omnibus (GSE63048). |

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
