## [Decision Letter]

Thank you for sending your work entitled “*In vivo* targeting of *de novo* DNA methylation by histone modifications in yeast and mouse” for consideration at *eLife*. Your article has been favorably evaluated by a Senior editor and three reviewers, one of whom, Bing Ren, is a member of our Board of Reviewing Editors.

The Reviewing editor and the other reviewers discussed their comments before we reached this decision, and the Reviewing editor has assembled the following comments to help you prepare a revised submission.

All reviewers agreed that your findings are important in understanding what determines the specificity of the DNA methylation machinery and that your experimental approach of expressing DNMT3b in an organism that lacks DNA methylation is unique and creative. Unfortunately, many key findings have just been published on January 21 in the journal *Nature* (2). However, since your manuscript was submitted before the publication of Baubec et al., the editors decided to proceed with the review process. In this case, we would like to ask you to revise your manuscript swiftly by addressing the comments below and return your revision within four weeks (the sooner the better), in order for us to further consider publication.

1) The authors have to provide a clear rationale for their selection of DNMT3b only (instead of DNMT3a or both) to study de novo DNA methylation, since DNMT3a has been shown to have stronger enzymatic activity than DNMT3b in (2), and DNMT3a (but not DNMT3b) is frequently mutated in tumors.

2) The authors need to discuss any differences of their work in comparison with [2].

3) The final section on mouse male germline DNA methylation is the least developed, but this should not prove difficult to improve. Many results and conclusions in the section “Correspondence between H3K36me3 and early DNA methylation in mammalian cells” are primarily supported by browser captures (Figure 6) rather than quantitative plots. This is not appropriate. Furthermore, there is a paucity of summary information on the WGBS experiments. An intelligent reader will refer to this in supplementary data in order to judge the reliability of the data.

---

## [Author Response]

*In this case, we would like to ask you to revise your manuscript swiftly by addressing the comments below and return your revision within four weeks of time (the sooner the better), in order for us to further consider publication*.

*1) The authors have to provide a clear rationale for their selection of DNMT3b only (instead of DNMT3a or both) to study de novo DNA methylation, since DNMT3a has been shown to have stronger enzymatic activity than DNMT3b in (*[2]*), and DNMT3a (but not DNMT3b) is frequently mutated in tumors*.

We expressed both murine de novo DNA methyltransferases in yeast, but we weren’t able to detect any expression both at RNA and protein levels of the murine DNMT3a in our transformed yeast strains. Moreover, whole genome bisulfite sequencing data of the DNMT3a strains do not show any level of DNA methylation. We do not understand the reason for why DNMT3a expression did not work, while DNMT3b did.

*2) The authors need to discuss any differences of their work in comparison with*
[2].

Discussion of similarities and differences have been included in the Introduction and Discussion sections of the manuscript, as illustrated below.

Introduction, fifth paragraph: “Recently it has been reported that the PWWP domain is important in specifying the localization of DNMT3b in mouse embryonic stem cells (2).”

Discussion, first paragraph: “It has been recently shown (2) that in embryonic stem cells the PWWP domain is responsible for the targeting of DNMT3b […] and during a period of biologically important de novo DNA methylation (germ cells).”

Discussion, second paragraph: “In our yeast system we detected an anti-correlation between transcript levels and DNA methylation […] into the gene-body, shaping their intragenic DNA methylation distribution.”

*3) The final section on mouse male germline DNA methylation is the least developed, but this should not prove difficult to improve. Many results and conclusions in the section* “*Correspondence between H3K36me3 and early DNA methylation in mammalian cells*” *are primarily supported by browser captures (*Figure 6*) rather than quantitative plots. This is not appropriate. Furthermore, there is a paucity of summary information on the WGBS experiments. An intelligent reader will refer to this in supplementary data in order to judge the reliability of the data*.

Figure 6 was modified: the previous genome browser snapshot (which was similar to that of 6D) was substituted with a quantitative plot showing that regions with high H3K36me3 levels at E13.5 are associated with the de novo DNA methylation increase occurring between E13.5 and E16.5. Data from WGBS experiments in mice ([37] and [43]) were summarized in Supplementary file 1 panel G and Supplementary file 2 panel C, in the same manner as the yeast DNA methylation data. Also, Figure 6 are quantitative metaplots over all protein coding genes, which show global correlation of DNA methylation and H3K36me3 (6F) and anti-correlation with H3K4me3 (6E).